# Local avalanche photodetectors driven by lightning-rod effect and surface plasmon excitations

Zhao Fu[1,2,7], Jia Liu[1,7], Meng Yuan[1], Jiafa Cai[1,3], Rongdun Hong[1,3], Xiaping Chen[1,3], Dingqu Lin[1,3], Shaoxiong Wu[1,3], Yuning Zhang[1], Zhengyun Wu[1,3], Zhanwei Shen [4]✉, Zhijie Wang [4]✉, Jicheng Wang [5]✉, Mingkun Zhang [1,6]✉, Zhilin Yang [1]✉, Deyi Fu [1]✉, Feng Zhang [1,3]✉ & Rong Zhang [1]✉

Sensitive avalanche photodetectors (APDs) that operate within the ultraviolet spectrum are critically required for applications in detecting fire and deep-space exploration. However, the development of such devices faces significant challenges, including high avalanche breakdown voltage, the necessity for complex quenching circuits, and thermal runaway associated with Geiger-mode avalanche operation. To mitigate these issues, we report on a 4H-SiC APD design utilizing micro-holes (MHs) structures and Al nano-triangles (NTs) to enhance surface electric field driven by strong localized surface plasmon excitations and lightning-rod effect. The device demonstrates a low avalanche breakdown voltage of approximately 14.5 V, a high detectivity of $2 \times 10^{13}$ Jones, a nanosecond-level response time, and repeated stable detections without the requirement of a quenching circuit. Collectively, when compared with the conventional wide-bandgap-based APDs, this device achieves a reduction in avalanche breakdown voltage by an order of magnitude. Consequently, the proposed APD configuration presents a promising candidate for ultraviolet detection and integrated optoelectronic circuits.

Avalanche photodetectors (APDs) are widely utilized in various applications, including lidar, chemical sensing, flame detection, ozone-hole sensing, and telecommunications, owing to their intrinsic ultra-high gain, which enables high sensitivity with a range extending from ultraviolet to terahertz[1–13]. To date, ultraviolet APDs based on wide-bandgap materials as SiC[14,15], GaN[16–21], $Ga_2O_3$[22,23], and ZnO[24,25] have shown impressive performance, achieving high gain (>1000) and high response speed (<1 μs). These capabilities are largely attributed to their inherent properties, including high avalanche characteristics, low

minority carrier lifetime, and intrinsic ultraviolet absorption[26]. Generally, conventional APDs typically operate in the Geiger mode under 95% of the critical electrical field to achieve high gain. However, several critical limitations arise when operating APDs in this mode. First, the avalanche state increases the risk of device breakdown, necessitating the use of quenching circuits to prevent permanent damage[14,15,27–33]. This requirement increases considerable complexity in their practical application. Second, the long-term stability and operational life of traditional APDs are compromised due to the stress induced by

[1]Department of physics, Xiamen university, Fujian, P. R. China. [2]College of Electrical Engineering, Tongling university, Anhui, P. R. China. [3]Jiujiang Research Institute of Xiamen University, Jiangxi, P. R. China. [4]Laboratory of Solid-State Optoelectronics Information Technology, Institute of Semiconductors, Chinese Academy of Sciences, Beijing, P. R. China. [5]School of Science, Jiangnan University, Jiangsu, China. [6]The Higher Educational Key Laboratory of Flexible Manufacturing Equipment Integration of Fujian Province, Xiamen Institute of Technology, Fujian, P. R. China. [7]These authors contributed equally: Zhao Fu, Jia Liu. ✉e-mail: zwshen@semi.ac.cn; wangzj@semi.ac.cn; jcwang@jiangnan.edu.cn; mkzhang@xmu.edu.cn; zlyang@xmu.edu.cn; dyfu@xmu.edu.cn; fzhang@xmu.edu.cn; rzhangxmu@xmu.edu.cn

continuous Geiger-mode operation[34]. Furthermore, wide-bandgap APDs functioning in Geiger mode require relatively high driving voltages (≥100 V), which makes them less suitable for applications that demand lower operational voltages. Therefore, it is critical to develop wide-bandgap APDs capable of achieving high gain at lower voltages for high stability, while eliminating the requirement for quenching circuits. Such advancements could significantly expand the range of APD applications.

An alternative approach to leveraging field enhancement is to construct semiconductor microstructures and metallic nanostructures, which have demonstrated considerable promise in enhancing local electric fields[35–40]. These enhancements arise from the accumulation of charges in the sharp zones of metallic nanostructures or near the corners of semiconductor microstructures, leading to localized amplification of the electric field[41]. However, there are relatively few reports on combining the lightning-rod effect with surface plasmon excitations to enhance the electric field to form an avalanche. By incorporating this field enhancement within semiconductor photodetectors, the internal electric field can reach avalanche field intensities (1–3 MV/cm) under lower reverse bias conditions.

In this study, a SiC-based APD design incorporating semiconductor microstructures and metallic nanostructures is presented and demonstrated with lightning-rod effect and surface plasmon excitation mechanisms. This design achieves a low breakdown voltage of 14.5 V with high detectivity of $2 \times 10^{13}$ Jones, high gain of $10^4$, and an improved response speed of nanoseconds. These advancements enable scalable and flexible platforms within mature SiC device technology, paving the way for the development of APDs capable of detecting weak signals without Geiger-mode operation. Moreover, the approach has broader implications, potentially influencing the design of wide-bandgap optoelectronic devices, quantum devices, and integrated optoelectronic circuits.

## Results

### Photoelectrical characterization

The structure of 4H–SiC avalanche photodetectors (APDs) with microhole (MH) and aluminum nano-triangles (Al NTs) was illustrated in Fig. 1. For specific details of fabrication, please refer to the Method section. Current-voltage (I–V) characteristics were measured to assess the performance of the proposed APDs. These measurements revealed that the local avalanche effect occurs at a reverse bias of approximately 14.5 V in devices with aluminum NTs in the SiC MHs, as depicted in Fig. 2a. The reduction in avalanche voltage offers significant advantages, particularly in preventing instantaneous breakdown and safeguarding the device from catastrophic damage. In addition, the photocurrent in devices incorporating Al NTs in the MHs was observed to be within the range of $10^{-5}$-$10^{-3}$ A for reverse biases between 20 V and 50 V, compared to devices without Al NTs, which exhibited photocurrents of approximately $10^{-9}$-$10^{-8}$ A. The gain of devices with Al NTs reached $10^4$ within a voltage range of a few tens of volts, whereas the gain in devices without Al NTs was significantly lower, only increasing by 6 times (Figs. 2a, S3). We also tested the dark currents and photocurrents of several devices with 4 μm, 8 μm and 10 μm MHs incorporating Al NTs and calculated their avalanche onset voltages and maximum avalanche gains, with error analysis histograms shown in Fig. 2b. The 4 μm MH devices exhibited smaller onset voltage fluctuations overall, while 8 μm and 10 μm devices showed larger fluctuations, primarily related to Al NTs distribution within the MHs and varying degrees of local avalanche enhancement. Additionally, despite gain fluctuations, 4 μm MH devices consistently achieved

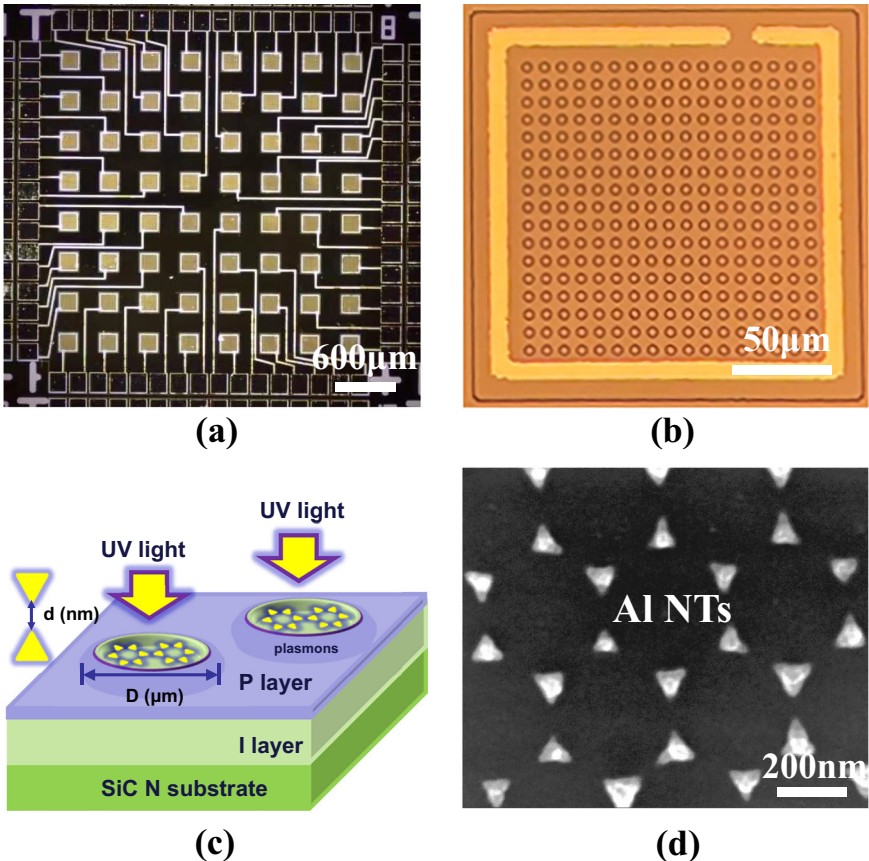

**Fig. 1 | Structural characterization and schematic diagram. a** Top-view of the whole 8×8 p-i-n APD array with MHs and Al NTs. **b** Single p-i-n APD pixel with MHs and Al NTs. **c** Magnified schematic of 4H–SiC p–i–n APD with MHs and Al NTs. **d** Section morphology and magnified view of Al NTs arrays in the MHs.

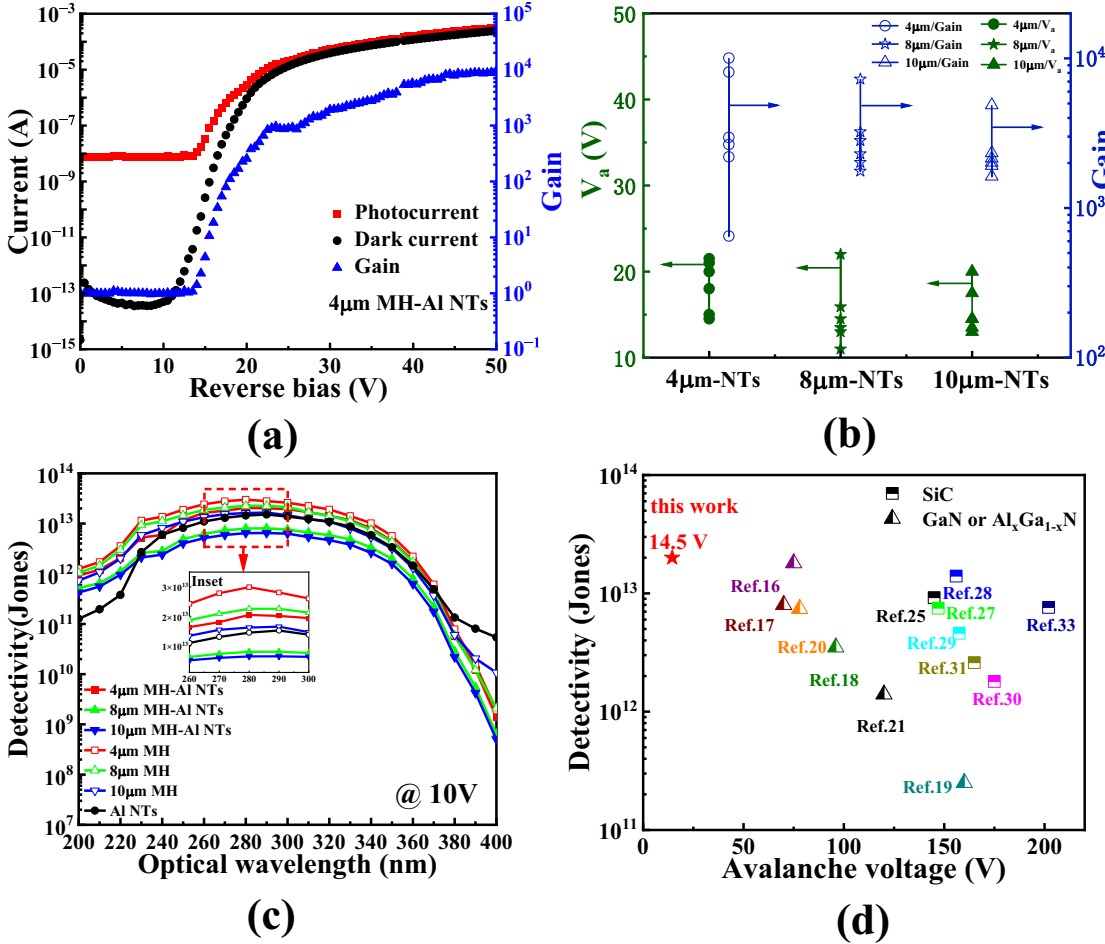

**Fig. 2 | Photoelectrical characterization and domestic-international comparison. a** The photocurrent, dark current, and gain versus voltage characteristics of the 4H−SiC p-i-n APDs with 4 μm MH and Al NTs. **b** Histograms of error analysis for the avalanche breakdown voltage (Va) and gain of APDs with Al NTs and MHs of different diameters (4 μm, 8 μm, and 10 μm). **c** The corresponding detectivity of APD devices with MHs without Al NTs, with MHs with Al NTs, and without MHs with Al NTs in the wavelength range of 200 nm to 400 nm at 10 V bias. The inset shows detectivity from 260 nm to 300 nm. **d** Comparison of detectivity at unity gain and avalanche breakdown voltage with other work (including 4H−SiC, GaN and $Al_xGa_{1-x}N$ APDs).

higher gains than 8 μm and 10 μm devices, reaching up to 10,556 times, mainly because their lateral region could be fully depleted, creating larger local avalanche regions. However, due to the inability to perfectly control the uniformity of the PS (polystyrene) sphere template arrangement, there may be some variability in the size and spacing of the Al NTs on each device's surface. Therefore, while each device experiences a local avalanche effect, not many avalanche devices can achieve a gain of $10^4$ as described in Fig. 2b. It can also be observed that not all 4 μm devices exhibit an avalanche onset voltage as low as 14.5 V, but they are all below the avalanche voltage of traditional APDs. This is primarily due to the non-uniform distribution of Al NTs. Similarly, the gain of the devices varies from $10^3$ to $10^4$. Detailed photocurrent, gain, and dark current for Al NT devices are provided in Figs. S2−S4.

From the spectral response data, the 4 μm MH device without Al NTs showed peak responsivity of 0.133 A/W at 280 nm at 10 V (58.9% External Quantum Efficiency (EQE)) as shown in Fig. S6, while the Al NT-equipped 4 μm MH device reached 0.129 A/W (57.2% EQE). 42.8% of the incident photons do not contribute to the generation of photogenerated carriers. This proportion accounts for light reflection, carrier recombination due to defects, and field enhancement. Reflection spectrum testing, as shown in Fig. S13, indicates that the reflectivity at 280 nm is approximately 16.0%. Therefore, the photons contributing to the field enhancement account for less than 26.8%. Similarly, 8 μm and 10 μm devices with Al NTs exhibited lower EQE values due to

partial light reflection by Al NTs. Compared to 4 μm devices, the reduced responsivity and EQE in larger MH devices stem from incomplete depletion of lateral electric fields, lowering carrier collection efficiency. Under Al NT and MH-induced local avalanche, the 4 μm MH device achieved $3.35 \times 10^{-4}$ A photocurrent with EQE of 2,374,473% at 50 V.

Furthermore, device detectivity was evaluated across wavelengths (Fig. 2c). The 4 μm MH devices with Al NTs achieved peak detectivity of $2.0 \times 10^{13}$ Jones at 280 nm under a reverse bias of 10 V. Notably, detectivity decreased with increasing MH size, which is attributed to indepletion region impairing carrier collection. Regardless of MH size, devices with Al NTs showed lower detectivity than counterparts without Al NTs, as Al NTs both reduce responsivity through light reflection and increase leakage current (Figs. S4, S6). As shown in Fig. S7, the Al NT devices in this work exhibit lower noise characteristics than conventional APDs due to localized avalanche effects.

A comparison of detectivity versus avalanche voltage was conducted to benchmark our devices against existing wide-bandgap semiconductor technology (Fig. 2d). Our APDs demonstrated superior performance, featuring the highest detectivity and the lowest avalanche voltage of 14.5 V. These characteristics underscore the exceptional performance metrics, including low avalanche voltage, high detectivity, and high gain. This balance is particularly advantageous in

the realm of micro- and nano-photonics, where surface plasmon excitations in MHs are leveraged in 4H–SiC APDs. Moreover, the stability of the device was evaluated to determine the impact of long-term operation on performance. After 13,000 h (1.5 years) of exposure, multiple I–V measurements showed consistency with the initial curve (Fig. S5). These results indicate that the proposed devices, incorporating Al NTs, exhibit robust and stable operation in the local avalanche state without the need for quenching circuits. In this study, high gain has been achieved in the detectors through stable and repeatable local avalanche at lower voltages, avoiding devastating bulk avalanche breakdown. This approach not only significantly reduces the avalanche voltage but also protects the devices from breakdown, which eliminates the need for a quenching circuit. The long-term stability and repeatability of the devices are demonstrated in Fig. S5.

## Lightning-rod effect and surface plasmon simulation

Our previous work investigated the impact of metal holes (MHs) on device performance and conducted a detailed analysis of how MH size affects device performance. Under the condition of maximizing the device's photocurrent, we determined the optimal MH size to be 4 μm. When the MH diameter is less than 4 μm, the photocurrent increases with the increase in MH size. However, when the MH size exceeds 4 μm, the photocurrent decreases with the increase in MH size[42,43]. In this study, we further investigated the individual impact of aluminum nanotubes (Al NTs) on device performance by characterizing the photocurrent, dark current, gain, and spectral response of devices incorporating Al NTs but no MHs (Figs. S2–S4, S6). Our results reveal that devices with Al NTs (but no MHs) exhibit no localized avalanche effect. The observed increase in both photocurrent and dark current primarily stems from the strong electric field at the Al NT tips, which effectively elevates the voltage applied to the device's $P^+$ layer, analogous to enhancing the reverse bias.

The electric field is a crucial factor in determining carrier collection and multiplication within APDs. Therefore, a theoretical simulation based on finite element analysis (FEA) was conducted using TCAD to investigate the electric field distribution in the absence of illumination (Fig. 3a, b). The bulk electric field distribution for the APDs with different MH sizes (Fig. 3a) indicates that a complete depletion layer is formed at the junctions for devices with 4 μm MHs and Al NTs. In contrast, the intrinsic i-layer of APDs with 8 μm and 10 μm MHs and Al NTs is only partially depleted. The undepleted regions serve as recombination centers, resulting in reduced carrier collection during avalanche events, which explains why the APDs with 4 μm MHs and Al NTs exhibit superior performance in terms of gain, detectivity, and avalanche voltage (Fig. 2b, c).

Certainly, the operational state of the device varies with different applied voltages, as illustrated in Fig. S8. Under a 5 V bias, the maximum internal electric field reaches 0.4 MV/cm, which is below the threshold for initiating local avalanche. At elevated voltages, the internal electric field intensifies further. With a 40 V reverse bias, the field strength attains 3 MV/cm, thereby extending the avalanche region and amplifying the avalanche effect, consequently increasing the current flow.

The local field enhancement induced by the lightning-rod effect near the corner of MH becomes prominent under different applied voltages. Under dark conditions, the local field enhancement at the edges of the MH, induced by the lightning-rod effect, becomes substantial at different applied voltages. For instance, at a reverse bias of 5 V, electric field aggregation is observed at the tips of the Al NTs near the electrode. However, the electric field intensity reaches only 0.74 MV/cm (Fig. 3b), which remains insufficient to initiate a local avalanche effect within the device. As the reverse bias increases to 15 V, the electric field intensity at the tips of the Al NTs rises to 2.28 MV/cm, causing the internal electric field of the device to reach 1.5 MV/cm (Fig. 3b), which is sufficient to induce a local avalanche effect. With

further increases in the reverse bias, the electric field intensity at the tips of the Al NTs continues to strengthen; at 40 V, it reaches a high value of 5.76 MV/cm, which leads to an expanded avalanche region and an increase in avalanche gain. Invariably, the regions where local avalanches occur remain concentrated at the edges of the MHs, as the electric field intensity at the Al NT tips diminishes with distance from the electrodes.

In contrast, for devices without Al NTs, the surface electric field intensity is approximately one-fifth of that observed in devices containing Al NTs. At a 15 V reverse bias, the surface electric field intensity at the edges of the MHs is only 0.41 MV/cm (Fig. S9), with the corresponding internal electric field within the device remaining below 0.4 MV/cm (Fig. S10), which is not adequate to trigger a local avalanche and yield higher gain. These field enhancement effects at the nanotube tips have been corroborated by other studies[41].

To further explore the local field enhancement mechanism of Al NTs under illumination, we employed the finite difference time domain (FDTD) method to simulate and validate the field intensity distribution of Al NTs. The spacing between Al NTs was set at 10 nm, 20 nm, 30 nm, and 80 nm, with the side length and thickness of the NTs set at 80 nm and 10 nm, respectively, in alignment with the experimental parameters. It was found that the spacing between adjacent NTs strongly influences the field enhancement (Fig. 3c). Specifically, when the spacing is reduced to less than 10 nm, the plasmon coupling field intensity increases by more than 30-fold. The strong confinement of the enhanced impact ionization at smaller gaps between NTs further extends the local avalanche region, enhancing the performance of the device at the local avalanche state. Although plasmonic coupling between bow-tie structures diminishes as the NT spacing increases from 20 nm to 80 nm, the field intensity remains more than 18 times greater than that of untreated tips. This field intensity is sufficient to cooperate with avalanche multiplication, confirming the effectiveness of the Al NTs surface plasmon excitations in coupling the high local electric field and improving device performance.

To further investigate the internal electric field distribution, we conducted COMSOL simulations on devices with varying Al NT spacings, as illustrated in Figs. S11, S12. The simulation results demonstrate unequivocally that devices incorporating Al NTs produce significantly stronger internal electric fields compared to Al-free devices, achieving a maximum field strength of 4.84 MV/cm at a spacing of 10 nm. Furthermore, the simulations reveal an inverse correlation between Al NT spacing and the maximum internal field strength. As the spacing increases, the field intensity diminishes. This phenomenon can be attributed to the enhanced coupling effects observed in closely spaced Al NTs, which serve to amplify the local electric field. These findings conclusively demonstrate the profound impact of Al NT spacing on device performance.

## Response time and band structure

The temporal spectral response of the 4H–SiC APDs was investigated using a 266 nm picosecond laser as the illumination source and an oscilloscope for detection. The rise time ($T_r$) and fall time ($T_f$) were defined as the times during which the impulse voltage rises from 10% to 90% and falls from 90% to 10% of the peak value, respectively. Figure 4a and its inset present the photocurrent response at an applied reverse bias of −10 V. The results demonstrate that the APD with Al NTs exhibits superior operational stability and reliability. Additionally, due to the p–i–n vertical structure and the inherently low minority carrier lifetime of the 4H–SiC material, the devices incorporating 4 μm MHs and Al NTs achieved a high response speed, with a $T_r$ of 2.2 ns and a $T_f$ of 4.2 ns (Fig. 4a inset). The $T_r$ of the proposed device represents an 82.2% reduction compared to the 12.4 ns response time reported for $Ga_2O_3$-based photodetectors[23]. Furthermore, it was observed that both devices, with and without Al NTs, exhibited longer response times at a

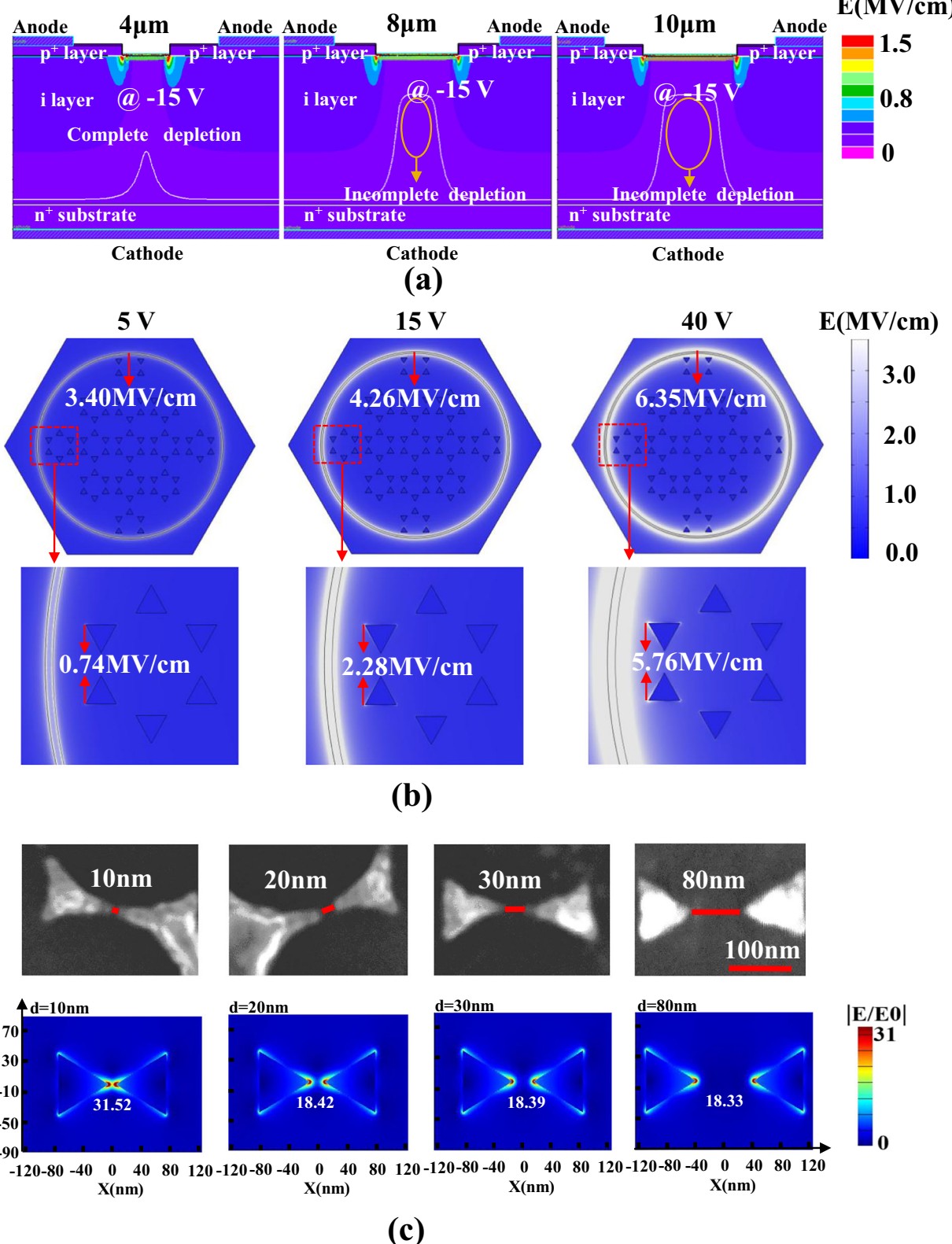

**Fig. 3 | Internal and surface plasmon tip-enhanced electric field simulation.**
**a** The internal electric field intensity of the devices with MH (4 μm, 8 μm and 10 μm) and Al NTs under the charge accumulation effect at the tips of Al NTs was simulated by applying 15 V reverse bias. **b** The electric field intensity at the tips of Al NTs was simulated under different reverse biases (5 V, 15 V, and 40 V), with the electric field intensities at the edge of the device electrodes and at the tips of the Al NTs being 3.4 MV/cm and 0.74 MV/cm, 4.26 MV/cm and 2.28 MV/cm, and 6.35 MV/cm and 5.76 MV/cm, respectively, under the reverse biases of 5 V, 15 V, and 40 V. **c** Through FDTD simulations, the field enhancement of Al NTs under illumination was investigated at various distances, corresponding to the interspacing of NTs in SEM images of 10 nm, 20 nm, 30 nm, and 80 nm. The field enhancement at the tips of the Al NTs was found to be 31.52, 18.42, 18.39, and 18.33, respectively, for interspaces of 10 nm, 20 nm, 30 nm, and 80 nm.

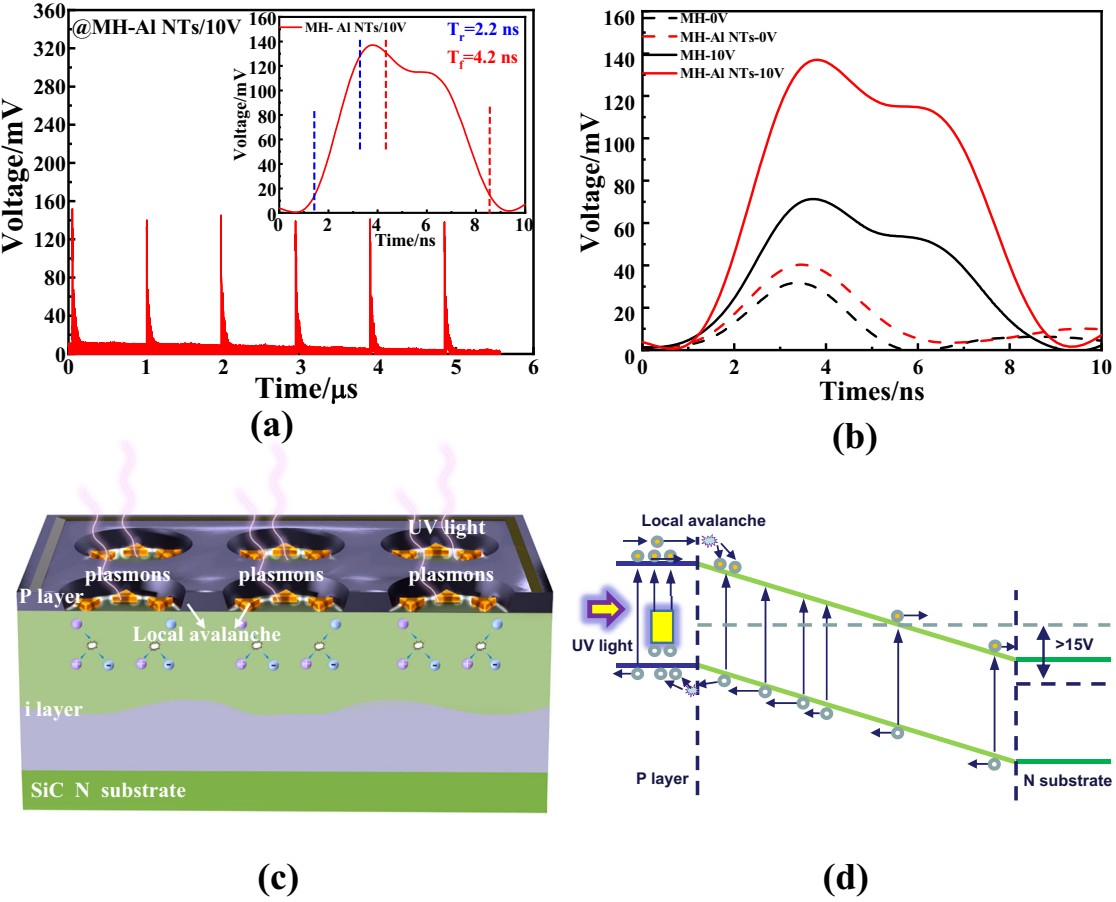

**Fig. 4 | Nanosecond-level response time and local avalanche schematic diagram. a** The response speed of the 4 μm-MH devices with Al NTs at 10 V, and the inset shows the single impulse response of the device with Al NTs at 10 V. **b** Comparisons of single impulses of the devices with and without Al NTs at 0 V and 10 V. **c** A schematic diagram of the structure that generates a local avalanche effect under illumination, excited by Al NTs. **d** Band diagram of the APD with MHs and Al NTs under the condition of illumination.

reverse bias of 10 V compared to 0 V (Fig. 4b, Fig. S14). Specifically, the $T_f$ of 4.2 ns at 10 V reverse bias was more than twice the 1.9 ns observed at 0 V. First, from the spectral response data, we observe that under 10 V reverse bias, the Al-containing device exhibits greater light reflection, resulting in lower photocurrent compared to the Al-free device. This suggests higher dynamic resistance in the Al-containing device. Following the RC oscillation circuit frequency formula ($f = 1/2\pi RC$), the reduced frequency (f) in Al-containing devices consequently yields longer response times ($T = 1/f$). Additionally, the frequency calculation reveals that total device capacitance impacts response speed. The Al-containing device demonstrates additional parallel capacitance between the device and Al nanoparticles relative to Al-free devices, increasing overall capacitance and consequently response time. Together, these factors account for the prolonged response times observed in Al-containing devices. Additionally, response speed may be influenced by the transit time of carriers, the drift current and the diffusion current within the device. These temporal performance enhancements highlight the strong carrier multiplication induced by the local field enhancement effect and demonstrate voltage-tunable response times in the proposed Al NT-integrated device. The combination of fast temporal response and low operational voltage positions the device as a promising candidate for high-speed ultraviolet (UV) detection applications.

The local avalanche mechanism and carrier transport in UV-illuminated devices are further elucidated through the schematic and band diagrams (Fig. 4c, d). The upper corner region of the device exhibits a higher electric field, corresponding to the local avalanche area. In this region, photogenerated carriers undergo collision ionization, resulting in the generation of electron-hole pairs. These pairs are efficiently separated by the reverse-biased electric field of the pn junction. In contrast to conventional APDs, the collision ionization process in the proposed device is localized to a specific area, while the remainder of the device remains stable. As a result, the photocurrent tends to saturate with increasing reverse bias, indicating that the device operates in a local avalanche state rather than a full avalanche state.

## Discussion

This study effectively enhanced the local electric field of 4H−SiC avalanche photodiodes featuring MHs by integrating the lightning rod effect with localized surface plasmon resonance of Al NTs. This approach achieved an unprecedentedly low avalanche initiation voltage of 14.5 V. Notably, these devices can operate in a local avalanche state without requiring quenching circuits, maintaining a high avalanche gain of $10^4$ and demonstrating excellent long-term stability.

The concept of local avalanche within 4H−SiC APDs introduces an innovative method for internal electric field regulation in semiconductor devices and holds potential for adaptation to other photodetectors based on wide-bandgap semiconductors. Future directions include optimizing the shape of metal nanostructures, with an emphasis on sharper features and optimized curvature radii, to maximize both the lightning rod effect and surface plasmon

resonance, thereby further improving device performance. These findings indicate that 4H−SiC materials and associated technologies constitute a robust and scalable platform for quantum-based light detection and imaging, offering ease of integration and high compatibility with industrial applications.

## Method

The 4H−SiC avalanche photodetectors (APDs) with micro-hole (MH) structures and aluminum nano-triangles (Al NTs) were fabricated into an 8 × 8 array with dimensions of 2 × 2 mm², as illustrated in Fig. 1a. The effective photosensitive area of a single pixel was designed to be 200 × 200 μm², as shown in Fig. 1b. The top surface of each device was etched with the MH structure, which serves as a window to enhance ultraviolet (UV) absorption. The MHs were etched through to the intrinsic i layer with diameters (D) of 4 μm, 8 μm, and 10 μm, respectively. As depicted schematically in Fig. 1c, the APD structure was processed using standard 4H−SiC epitaxial techniques. It consists of a 200 nm thick p-type layer ($N_A = 1 \times 10^{19}$ cm$^{-3}$), a 3000 nm thick i layer ($N_D = 1 \times 10^{15}$ cm$^{-3}$) on a 365 μm thick n substrate ($N_D = 5 \times 10^{18}$ cm$^{-3}$). Notably, in contrast to conventional APD structures, the i layer of the photodiode in this design is directly exposed to UV light, enabling enhanced UV detection capabilities. The proposed APD design incorporates an array of Al hexagonal NTs within the MHs, positioned on the 4H−SiC i-layer, as shown in Fig. 1d. The diameter (D) of the MHs and the spacing (d) between the NTs were varied from 4 μm to 10 μm and 10 nm to 80 nm, respectively. The Al NTs were designed with a side length of 80 nm and a thickness of 10 nm. The geometric features of the Al NTs could be precisely controlled using a polystyrene (PS) microsphere template[44,45]. Further details regarding the fabrication and measurement process are provided as follows.

### Devices fabrication

The local avalanche 4H−SiC PDs were fabricated and processed as follows. Firstly, a beveled mesa of the device was processed by photoresist reflow technology to effectively suppress the edge breakdown of the device. Then, a standard photolithography process was applied to obtain the MH array patterns by using AZ5214E positive photoresist. Subsequently, the inductive coupled plasma (ICP) etching was performed with a gas flow rate of 30 sccm (CF$_4$)/5 sccm (O$_2$), MHs with different diameters (4 μm, 8 μm and 10 μm) were etched and the MH spacing was fixed to 5 μm. By controlling the etching time, MH with a depth of 220 nm was obtained using an atomic force microscope (AFM). It ensures that the MHs were etched to the i layer, which allows more UV light to be absorbed directly by the i layer. Then, a thermally grown SiO$_2$ with a thickness of 45 nm was grown on the surface to restrict the leakage current of the device and suppress the edge breakdown of MHs. Ohmic contact metals on the p$^+$ layer and the backside of n$^+$ substrate was obtained by Ti/Al/Ti/Au (40/60/10/100 nm) and Ti/Ni/Ti/Au (5/200/5/150 nm), respectively, followed by a rapid thermal annealing process of 1000 °C/2 min in Ar ambient. Then, Ti/Au (20/300 nm) were sputtered as wire-bonding pads.

### Al nano-triangles (NTs) fabrication

First, the polystyrene (PS) microspheres were mixed with ethanol in a 1:1 ratio and ultrasounded for 10 min. Secondly, the prepared devices were placed in deionized water, and the water surface was just over the sample, and then the PS microspheres were slowly injected into the surface of the deionized water through the syringe until the PS microspheres were just covered the entire water surface. The deionized water was then slowly pumped down, and the PS microspheres were spread all over the devices surface (Fig. S1a). Finally, a 10 nm thick Al film was prepared by magnetron sputtering method, which was then immersed in tetrahydrofuran to remove the PS microspheres template and form regular Al NTs (Fig. S1b).

### Photoelectrical performance measurement

Figure S15a shows a schematic diagram of the device performance measurement system described in the manuscript. The system primarily consists of two integrated parts: the optical path and the electrical circuit. The optical path, which generates and conditions the test light, is composed of a xenon lamp, an optical fiber, and a monochromator. The electrical measurement section is contained within an electromagnetic shielding box and incorporates a probe station connected to a 4200A-SCS semiconductor parameter analyser. The current-voltage (I−V) characteristics of the semiconductor devices were measured using the source measure units (SMUs) of the analyser.

Prior to device testing, the intensity of the ultraviolet light emitted from the xenon lamp was calibrated. This calibration was performed by integrating a UV-enhanced Si-222 standard photodetector into the system. The UV light signal, after being conditioned by an adjustment objective lens, was focused onto the active surface of the standard photodetector to generate a photocurrent. The calibrated optical power values across the UV wavelength range were then calculated from the known spectral responsivity of the standard detector.

## Data availability

The main data supporting the findings of this study are available within the article and its Supplementary Figures. The source data underlying Figs. 2, 4, Supplementary Figs. S2−S7 and S13−S14 are provided as a Source Data file. The data that support the plots within this paper and other findings of this study are available from the corresponding author upon reasonable request. Source data are provided with this paper.

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

## Acknowledgments

This work was supported by the National Natural Science Foundation of China (Grant No. 62274137 and 62174143), National Key Research and Development Program of China (Grant No. 2023YFB3609500, Grant No.2023YFB3609502), Jiangxi Provincial Natural Science Foundation (Grant No. 20232BAB202043), and the Science and Technology Project of Fujian Province of China (Grant No. 2020I0001).

## Author contributions

The work was designed by Z.F. and F.Z. and supervised by Z.S., Z.J.W., J.W., M.Z., Z.Y., D.F., F.Z., and R.Z. The device was fabricated by Z.F., J.L., and M.Y. D.L., X.C., S.W., and R.H. contributed to the photoelectrical performance measurement. Z.F., J.L., M.Y., and J.C. helped with the structure simulation and mechanism explanation. Z.F., Y.Z., and Z.Y.W. participated in the response time characterization and band structure interpretation. The draft was written by Z.F. and J.L. The manuscript was modified by Z.F., F.Z., and J.L. All authors discussed and commented on the manuscript.

## Competing interests

The authors declare no competing interests.
