## [Transparent Peer Review file · Nature Communications]

Local Avalanche Photodetectors Driven by Lightning-rod Effect and Surface Plasmon Excitations

Corresponding Author: Professor Feng Zhang

Version 0:

Reviewer comments:

Reviewer #1

(Remarks to the Author)

Local Avalanche Photodetectors Driven by Lightning Rod Effect and Surface Plasmon Excitations by Fu et al.
The work is interesting and describes the impact of the Lightning Rod Effect and Surface Plasmon Excitations on 4H-SiC; however, several points require the author's attention for improvement:

1. Why are the detectivity values very similar at 270 nm excitation for a -10V bias in the 4 μm MH-AI-NTs, while there is significant variation for the 8 μm and 10 μm samples?
2. What is the noise level in the fabricated device?
3. The charge accumulation effect during the AI-NT process was simulated only at a 15V reverse bias. What is the impact of varying the reverse bias?
4. Strong carrier multiplication has been reported. The external quantum efficiency (EQE) should be included in the results and discussed.

Reviewer #2

(Remarks to the Author)

This manuscript presents a novel 4H-SiC avalanche photodetector (APD) designed for ultraviolet detection, addressing key challenges such as high breakdown voltage and the need for complex quenching circuits. By incorporating micro-hole (MH) structures and aluminum nano-triangles (NTs), the device enhances the surface electric field through localized surface plasmon excitations and the lightning-rod effect.

While the idea is novel and of interest for the community, the way that the data is currently presented is quite misleading and there are a few technical issues that I would like to point out: I recommend this paper to undergo major revision before being fit for publication.

- in general, the authors fail to clearly explain the different effects of the two design strategies they employed: MH and AI NT. There is no clear discussion of the effect of one vs the other, and most results involve the simultaneous implementation of both strategies. For example, Fig 2c shows a negligible improvement in detectivity granted by the inclusion of AI NT, contrary to what the authors claim across the manuscript. This is perhaps because the AI NT increase both the photocurrent and the dark current simultaneously, as noted at line 162-164, with opposing effects on the detectivity. I believe that the addition of a detailed and quantitative discussion of each the individual effect of each of the two strategies needs to be added to the manuscript: this can be done for example by merging some of the plots in Fig. 2a and S2.

- Similarly to what noted above, there seems to be no experimental characterization of what effect the spacing between AI NT has on performance, as hinted by the simulations (line 194).

- I have a particular issue with Fig 2c and 2d. The specific detectivity was invented to express the quality of a specific bulk material platform for photodetection, and cannot be freely applied to dimensionally-structured materials without specific caveats [Wang, F., Zhang, T., Xie, R. et al. How to characterize figures of merit of two-dimensional photodetectors. Nat Commun 14, 2224 (2023)]. The authors fail to mention what exact numbers were used to calculate the detectivity, in terms of responsivity, dark current and area. In particular, I suspect that the authors assumed an extremely small photosensitive area in their calculations, which is a conceptual mistake, as this is equivalent to discarding most of the incident light to be detected. In my opinion, the area used here should be the pitch of one square pixel in Fig. 1a. This will give a real estimation

of how good the array is at detecting incident light (from the reference above: “for plane array devices, the entire device tiling the plane needs to be regarded as the optical area”). Otherwise, the authors may use a smaller area if they demonstrate a measurement of the performance of the array with a microns array, which is capable of focusing the incident light onto an area smaller than the pixel size. I suggest that the authors specify the numbers used in their calculation, and add other performance metrics, such as internal and external quantum efficiency (the latter based on the total pixel area), which give a better indication of the useful performance of the detector. This is particularly important as plasmonic enhancement is known to introduce additional loss [Khurgin, J. How to deal with the loss in plasmonics and metamaterials. Nature Nanotech 10, 2–6 (2015)], and it’s necessary to estimate whether this strategy is worthwhile and provides a net improvement despite the plasmonic loss.

- the authors should provide a clearer explanation of why the presented architecture allows to avoid the need for quenching circuits: is this because the devices are operated at sub-breakdown voltages? If this is the case, what is the advantage compared to operating conventional detectors at sub-breakdown voltages (say devices without AI NT at 50 V)? The sentence in lines 113-115 “The reduction in avalanche voltage offers significant advantages, particularly in preventing instantaneous breakdown and safeguarding the device from catastrophic damage” is unclear or misleading: if the avalanche voltage is lower, it is typically easier to trigger breakdown and not harder.

- the schematic in Fig. 4d is a simple representation of any avalanche process, without and specific relation to the presented devices (e.g. local avalanche, field enhancement), and I believe fails to capture the complex physics in action in these devices. In fact, the incident photons that provide a local field enhancement as shown in Fig. 3c do not contribute to the photo carrier generation. To investigate this, one should put a monitor plane in the FDTD simulation, situated a few 10s nm inside the substrate underneath the AI NT, and compare the transmitted number of photons vs the incident ones: this portion is the one contributing to photo carrier generation. The authors should implement and discuss this model.

- The sentence at lines 217-220 “This increase is attributed to the acceleration of photogenerated carriers within the space-charge region at higher reverse bias, necessitating a longer response time. Moreover, both T_r and T_f in devices with AI NTs increased by approximately 10% compared to devices without AI NTs” sounds counter-intuitive, as the higher voltage will accelerate charges faster. In fact, I suspect this increase in response time is due to the higher current changing the dynamic resistance and hence the RC of the device, as evidenced by the fact that the same increase is also registered for devices with AI NT at the same voltages. If the authors disagree, they should provide an objective argument for their claims.

- finally, I believe the quality and presentation of the figures graphics can be significantly improved: for starter, the largest fonts in several figures are around ten times larger than the smallest in the same figures. Fig. 1a and 1b lack scale bars, and some of the numbers in Fig. 3 are not explained in the caption. The caption of Fig 2b mentions “statistical comparison”, but no statistics is shown or mentioned in the figure itself: I guess the bar plots are the average of some number of devices, but can the authors specify how many devices, how is this average obtained and insert the corresponding error bars? These are the minimum necessary improvements, but there are several more that can be implemented.

Version 1:

Reviewer comments:

Reviewer #1

(Remarks to the Author)

Local Avalanche Photodetectors Driven by the Lightning-Rod Effect and Surface Plasmon Excitations by Fu et al.

The authors have comprehensively addressed all the concerns raised by the reviewers. In light of this, the manuscript may be accepted for publication.

Reviewer #2

(Remarks to the Author)

I commend the efforts of the authors and recognize that the quality of the manuscript has substantially improved. However, there is still a key unsolved issue, which I believe to be fundamental to the validity of the manuscript:

- the main issue is with the answer to point 3 of reviewer#2 which is incomplete at best: the calculations shown are still missing a key factor, which is the incident light power that is used in calculating the responsivity values. Just extracting from the answer to point 5, 58.9 nW is used: is this power entirely focused inside the 4- μm MH or over the entire 200x200 μm pixel? I suspect that this value is obtained from the light incident over a large area of a few mm beam size, normalized to the area of the 4- μm MH ($12.6 \cdot 10^{-12} \text{ m}^2$), which is consistent with this number being later used for the EQE calculation. This is a wrong procedure because this area is not the same utilized in the detectivity calculation (i.e. $200 \times 200 \mu\text{m}^2$, as mentioned by the authors). Since the authors do not provide any details on the experimental setup, it is hard to infer what numbers were used in the calculations. The best practice would be to: 1. Describe the setup (e.g. “a source with ___ mW total power was focused on a ___ m^2 area, resulting in a power ___ W incident over the $200 \times 200 \mu\text{m}^2$ pixel area”); 2. Report the numbers used for both the responsivity and the detectivity calculations.

Other comments:

- the discussion and calculations with specific numbers used in the answers to point 2 of reviewer#1 and point 3 and 5 of reviewer#2 should be added to the supplementary

- the following paragraph on the statistics in figure 2(b) should be added to the text or supplementary “To be honest, due to the inability to perfectly control the uniformity of the PS (polystyrene) sphere template arrangement, there may be some variability in the size and spacing of the AI NTs on each device's surface. Therefore, while each device experiences a local avalanche effect, not many avalanche devices can achieve a gain of 10^4 . For devices with different MH sizes, we tested the photocurrent and dark current of over 20 individual devices, and an error analysis histogram is presented in Fig. 2(b). It

can be observed that not all 4 μm devices exhibit an avalanche onset voltage as low as 14.5 V, but they are all below the avalanche voltage of traditional APDs. This is primarily due to the non-uniform distribution of AI NTs. Similarly, the gain of the devices varies from 10^3 to 10^4 .”

Version 2:

Reviewer comments:

Reviewer #2

(Remarks to the Author)

The authors have addressed all points raised and concerns, and I believe the clarity and quality of the manuscript to be substantially improved: I recommend the manuscript for publication.

Dear Reviewers:

Thank you for your letter and the reviewers' comments on our submitted manuscript entitled "Local Avalanche Photodetectors Driven by Lightning-rod Effect and Surface Plasmon Excitations". These comments are very constructive and helpful to our paper and future research. We have studied comments carefully and have made corrections which we hope meet with approval. The detail responses to the reviewers' comments are as follows:

Reviewer #1 (Remarks to the Author):

Local Avalanche Photodetectors Driven by Lightning Rod Effect and Surface Plasmon Excitations by Fu et al. The work is interesting and describes the impact of the Lightning Rod Effect and Surface Plasmon Excitations on 4H-SiC; however, several points require the author's attention for improvement:

1. Why are the detectivity values very similar at 270 nm excitation for a -10V bias in the 4 μm MH-Al-NTs, while there is significant variation for the 8 μm and 10 μm samples?

Answer: Thanks for your comments. We have recalculated detectivity based on the measured dark current and spectral responsivity at a reverse bias of 10 V. The measured dark currents are a little bit higher than that at a bias less than 10 V. The detectivity curves have been corrected based on the higher dark current and spectral responsivity as follows:

Fig. 2 (c) The corresponding detectivity with the wavelength range from 200 nm to 400 nm for the APDs with and without Al NTs at 10 V reverse bias. The inset shows detectivity from 260 nm to 300 nm.

Fig.S4 and S6: (a) Spectral response and quantum efficiency graphs of devices with different MH diameters (4 μm, 8 μm, and 10 μm). (b) Spectral response and quantum efficiency graphs of devices with different MH diameters (4 μm, 8 μm, and 10 μm) with Al NTs under 10 V reverse bias. (c) The comparison of dark current for the PDs without Al NTs for different diameters MHs (4 μm, 8 μm and 10 μm). (d) The comparison of dark current for the PDs with Al NTs for different diameters MHs (4 μm, 8 μm and 10 μm).

First, the calculation formula for the normalized detectivity (D^*) $D^* = R \times \sqrt{\frac{A}{2qI_d}}$ indicates that the detectivity mainly depends on the responsivity and the dark current density. According to Figure S4 and Figure S6, the detectivities of the devices with different MH sizes under conditions with and without aluminum (Al) has been obtained based on the responsivities and dark currents.

As illustrated in the attached Figure S8, the 4 μm devices exhibit full depletion of the lateral electric field, whereas the 8 μm and 10 μm devices do not achieve complete depletion. This results in relatively lower responsivity and detectivity for the 8 μm and 10 μm devices. From the perspective of dark current, as shown in Figure S4, the full depletion of the lateral electric field in the 4 μm devices effectively suppresses both surface and bulk dark currents. In contrast, for the 8 μm and 10 μm devices, the

incomplete depletion of the electric field leads to increased bulk diffusion current and surface recombination current as the bias voltage increases.

Furthermore, irrespective of the device's MH size, it is evident that the devices without Al exhibit higher responsivity and detectivity compared to those with Al. This is primarily because localized avalanche breakdown has not yet occurred at a reverse bias of 10 V. The presence of Al nanoparticles (Al NTs) increases light reflection to some extent, thereby reducing the incidence efficiency and leading to a decrease in photocurrent. Consequently, the responsivity and detectivity are also reduced. These findings have been discussed in the main text and highlighted accordingly.

According to your comment, we have revised the main text of this section on **page 6** of the manuscript. The revised part is as follow:

“Furthermore, device detectivity was evaluated across wavelengths (Fig.2(c)). The 4 μm MH devices with Al NTs achieved peak detectivity of 2.0×10^{13} Jones at 280 nm under a reverse bias of 10 V. Notably, detectivity decreased with increasing MH size, which is attributed to indepletion region impairing carrier collection. Regardless of MH size, devices with Al NTs showed lower detectivity than counterparts without Al NTs, as Al NTs both reduce responsivity through light reflection and increase leakage current (Figs. S4, S6).”

2. What is the noise level in the fabricated device?

Answer: The avalanche noise level depends on the ionization rate ratio (α_N/α_P) and the avalanche gain (M). For the case of hole injection, the noise level can be expressed as :

$$F = M \left[1 - (1 - k) \left(\frac{M - 1}{M} \right)^2 \right]$$

$$\approx kM + (2 - \frac{1}{M})(1 - k)$$

where $k = \alpha_N/\alpha_P$, which remains constant throughout the avalanche region. For SiC material, the impact ionization coefficients of electrons and holes can be expressed as:

$$\alpha_N(E) = 1.69 \times 10^6 \text{ cm}^{-1} \exp \left[- \left(\frac{9.96 \times 10^6 \text{ V/cm}}{E} \right)^{1.6} \right] \quad (\text{electron})$$

$$\alpha_P(E) = 3.32 \times 10^6 \text{ cm}^{-1} \exp \left[- \left(\frac{1.07 \times 10^7 \text{ V/cm}}{E} \right)^{1.1} \right] \quad (\text{hole})$$

where E represents the electric field intensity.

The avalanche electric field intensity in conventional SiC APDs is typically 3 MV/cm. By substituting $E = 3 \text{ MV/cm}$ into the equations above, we calculate that $k \approx 0.032$. Consequently, the noise level in conventional devices can be expressed as:

$$F = 0.032M - 0.968/M + 1.936$$

For comparison, the local avalanche electric field intensity in our fabricated device is 1.5 MV/cm under a reverse bias of 15 V as shown in Fig. S8. We therefore simulated the local avalanche electric field intensity under different reverse bias conditions. Since the gain is related to the reverse bias (as shown in Fig. 2a), Fig. S7 was obtained by curve fitting.

Fig.S7 Comparison of excess noise factor in the device without MH-AI NTs and the fabricated device.

The results show that the noise level increases with avalanche gain. The fabricated device exhibits a lower noise level compared to the device without MH-AI NTs, owing to localized avalanche effects.

According to your comment, we have revised the main text of this section on **page 7** of the manuscript. The revised part is as follow:

“As shown in Fig. S7, the AI NT devices in this work exhibit lower noise characteristics than conventional APDs due to localized avalanche effects.”

3.The charge accumulation effect during the Al-Nt process was simulated only at a 15V reverse bias. What is the impact of varying the reverse bias?

Answer: The application of varying reverse biases to the AI NTs device significantly alters both the internal electric field distribution and maximum field strength. As discussed in the main text, we have conducted a comprehensive analysis of how different reverse biases affect charge accumulation. Figure S8 in the supplementary materials presents detailed electric field distribution diagrams under 5 V, 15V and 40 V reverse biases, while the main text focuses on the internal electric field distribution at 15 V reverse bias. At this bias voltage, the electric field strength at the edge of the MH structures (MHs) reaches 1.5 MV/cm, sufficient to trigger local avalanche phenomena.

In comparison, Figure S8 demonstrates that a 5 V bias generates a maximum internal electric field of only 0.4 MV/cm, below the threshold for local avalanche initiation. As the bias voltage increases, the internal electric field intensifies proportionally reaching 3 MV/cm at 40 V reverse bias. This stronger field expands the avalanche region and enhances the avalanche effect, consequently increasing the current flow. Our static-field simulations under various conditions (Fig. 3(b)) confirm that higher applied voltages produce stronger electric fields, thereby increasing the probability of local avalanche occurrences.

According to your comment, we have revised the main text of this section on **page**

9 of the manuscript. The revised part is as follow:

“Certainly, the operational state of the device varies with different applied voltages, as illustrated in Figure S8. Under a 5 V bias, the maximum internal electric field reaches 0.4 MV/cm, which is below the threshold for initiating local avalanche. At elevated voltages, the internal electric field intensifies further. With a 40 V reverse bias, the field strength attains 3 MV/cm, thereby extending the avalanche region and amplifying the avalanche effect, consequently increasing the current flow.”

Fig.S8. The internal electric field intensity of the devices with MH (4 μm , 8 μm and 10 μm) and Al NTs under the charge accumulation effect at the tips of Al NTs was simulated by applying (a) 5 V, (b) 15 V, (c) 40 V reverse bias.

4. Strong carrier multiplication has been reported. The external quantum efficiency (EQE) should be included in the results and discussed.

Answer: Thank you for your comment. We will include the spectral data for the device's responsivity and quantum efficiency before local avalanche occurs in the supplement. The spectral response data show that the 4 μm micro-hole device without Al NTs achieves the highest responsivity of 0.133 A/W at 280 nm at 10V, corresponding to an external quantum efficiency of 58.9% as shown in Fig. S6. Meanwhile, the 4 μm MH device with Al NTs shows a maximum responsivity of 0.129 A/W at 280 nm at 10V, with a corresponding external quantum efficiency of 57.2%. Similarly, the 8 μm and 10 μm devices with Al NTs also exhibit lower external quantum efficiency values. This reduction occurs because the aluminum metal partially reflects incident light,

leading to decreased photocurrent. Compared to the 4 μm devices, the lower responsivity and external quantum efficiency in the 8 μm and 10 μm devices result from their lateral electric fields not being fully depleted, which reduces carrier collection efficiency.

When local avalanche is triggered at 50V by the Al NTs and MHs under light excitation, the 4 μm MH detectors demonstrates significantly enhanced performance with photocurrent reaching 3.35×10^{-4} A. This corresponds to a responsivity of 5357.4 A/W and a quantum efficiency of 2,374,473%. We have described these changes in external quantum efficiency in the main text and highlighted these results.

According to your comment, we have revised the main text of this section on **page 6** of the manuscript. The revised part is as follow:

“From the spectral response data, the 4 μm MH device without Al NTs showed peak responsivity of 0.133 A/W at 280 nm at 10 V (58.9% External Quantum Efficiency (EQE)) as shown in Fig. S6, while the Al NT-equipped 4 μm MH device reached 0.129 A/W (57.2% EQE). 42.8% of the incident photons do not contribute to the generation of photogenerated carriers. This proportion accounts for light reflection, carrier recombination due to defects, and field enhancement. Reflection spectrum testing, as shown in Fig. S13, indicates that the reflectivity at 280 nm is approximately 16.0%. Therefore, the photons contributing to field enhancement account for less than 26.8%. Similarly, 8 μm and 10 μm devices with Al NTs exhibited lower EQE values due to partial light reflection by Al NTs. Compared to 4 μm devices, the reduced responsivity and EQE in larger MH devices stem from incomplete depletion of lateral electric fields, lowering carrier collection efficiency. Under Al NT and MH-induced local avalanche, the 4 μm MH device achieved 3.35×10^{-4} A photocurrent with EQE of 2,374,473% at 50 V.”

Reviewer #2 (Remarks to the Author):

This manuscript presents a novel 4H-SiC avalanche photodetector (APD) designed for ultraviolet detection, addressing key challenges such as high breakdown voltage and the need for complex quenching circuits. By incorporating micro-hole (MH) structures and aluminum nano-triangles (NTs), the device enhances the surface electric field through localized surface plasmon excitations and the lightning-rod effect.

While the idea is novel and of interest for the community, the way that the data is currently presented is quite misleading and there are a few technical issues that I would like to point out: I recommend this paper to undergo major revision before being fit for publication.

1. in general, the authors fail to clearly explain the different effects of the two design strategies they employed: MH and Al NT. There is no clear discussion of the effect of one vs the other, and most results involve the simultaneous implementation of both strategies. For example, Fig 2c shows a negligible improvement in detectivity granted by the inclusion of Al NT, contrary to what the authors claim across the manuscript. This is perhaps because the Al NT increase both the photocurrent and the dark current simultaneously, as noted at line 162-164, with opposing effects on the detectivity. I believe that the addition of a detailed and quantitative discussion of each the individual effect of each of the two strategies needs to be added to the manuscript: this can be done for example by merging some of the plots in Fig. 2a and S3.

Answer: Thank you for your comment. Firstly, I would like to address your concern regarding the lack of separate discussion about the impact of MH and Al NTs on device performance in the article. The influence of MH on the performance of vertical p-i-n devices has been thoroughly analyzed in our previous work, as referenced in the paper "Local Avalanche Effect of 4H-SiC p-i-n Ultraviolet Photodiodes With Periodic Micro-Hole Arrays"(ref.44), which we have now included in the references section of the article.

The metal-dot (MH) structure was designed to mitigate the ineffective absorption of ultraviolet light by the P+ layer, thereby increasing the effective incident ultraviolet light and enhancing both the responsivity and quantum efficiency of the device. Theoretically, larger MHs would allow for better ultraviolet light absorption. However, in practice, larger MHs result in the partial depletion of the i-layer at the MH sites, which hinders the collection of photogenerated carriers and leads to a lower photocurrent. Therefore, to maximize the photocurrent of the devices, we determined the optimal MH size to be 4 μm . When the MH diameter is less than 4 μm , the photocurrent increases with the MH size. However, when the MH size exceeds 4 μm , the photocurrent decreases as the MH size increases (ref. 44). Moreover, electric field concentration occurs at the edges of the MHs, leading to a minor localized avalanche phenomenon at a reverse bias of 40 V, with a gain of up to 5 times. However, the obtained avalanche breakdown voltage remains relatively high, and the avalanche gain is small, which does not meet current demands for weak signal detection.

To further explore the effects of aluminum nanotubes (Al NTs), we fabricated devices without MHs but incorporated Al NTs, creating a contrast with devices that only have MHs and those that have both MHs and Al NTs. We tested the photocurrent, dark current, gain, and spectral response of the devices without MHs but with Al NTs, as shown in Figs. S2, S3, S4, and S6. Our results indicate that for devices without MHs but with Al NTs, there was no localized avalanche effect. While both the photocurrent and dark current increased, this was primarily due to the high electric field at the tips of the Al NTs, which applied a high voltage to the P⁺ layer of the device, similar to the effect of increasing the reverse bias.

In this paper, we further investigate the application of MHs, as well as their surface plasmonic resonance properties and lightning rod effect, to enhance the localized avalanche of the device, enabling higher avalanche gain at lower biases. In summary, this study primarily focuses on the impact of Al NTs on device performance based on the MH structure. We have discussed the influence of MHs on device performance in the main text and highlighted this discussion accordingly. Regarding your suggestion to merge Figure 2(a) and Figure S3, we selected the best-performing 4 μm devices from different types of devices as representatives to showcase their photocurrent, dark current, and gain.

According to your comment, we have revised the main text of this section on **page 9** of the manuscript. The revised part is as follow:

“Our previous work investigated the impact of metal holes (MHs) on device performance and conducted a detailed analysis of how MH size affects device performance. Under the condition of maximizing the device's photocurrent, we determined the optimal MH size to be 4 μm. When the MH diameter is less than 4 μm, the photocurrent increases with the increase in MH size. However, when the MH size exceeds 4 μm, the photocurrent decreases with the increase in MH size^{44,45}. In this study, we further investigated the individual impact of aluminum nanotubes (Al NTs) on device performance by characterizing the photocurrent, dark current, gain, and spectral response of devices incorporating Al NTs but no MHs (Figs. S2–S4, S6). Our results reveal that devices with Al NTs (but no MHs) exhibit no localized avalanche effect. The observed increase in both photocurrent and dark current primarily stems from the strong electric field at the Al NT tips, which effectively elevates the voltage applied to the device's P⁺ layer analogous to enhancing the reverse bias.

The electric field is a crucial factor in determining carrier collection and multiplication within APDs. Therefore, a theoretical simulation based on finite element analysis (FEA) was conducted using TCAD to investigate the electric field distribution in the absence of illumination (Figs. 3(a) and 3(b)). The bulk electric field distribution for the APDs with different MH sizes (Fig. 3(a)) indicates that a complete depletion layer is formed at the junctions for devices with 4 μm MHs and Al NTs. In contrast, the intrinsic i-layer of APDs with 8 μm and 10 μm MHs and Al NTs is only partially depleted. The undepleted regions serve as recombination centers, resulting in reduced carrier collection during avalanche events, which explains why the APDs with 4 μm MHs and Al NTs exhibit superior performance in terms of gain, detectivity, and avalanche voltage (Figs. 2(b) and 2(c)).”

2. Similarly to what noted above, there seems to be no experimental characterization of what effect the spacing between Al NT has on performance, as hinted by the simulations (line 194).

Answer: Thank you for your comment. Controlling the spacing of Al NTs is indeed a sapiential idea to study their impact on the avalanche effect. However, there are significant challenges in achieving uniformly small spacing of Al NTs (several nanometers) on the device at the moment.

First, the designed spacing of Al NTs in this study is approximately 80 nm, achieved using polystyrene (PS) beads with a diameter of 260 nm, representing the smallest size we can currently achieve with high reproducibility. To obtain uniform spacing below 80 nm, smaller PS beads would be required. However, arranging such smaller beads into a single-layer close-packed template poses significant challenges and remains practically unattainable at this stage.

Second, due to the difficulty in achieving perfect uniformity when arranging PS beads into a single-layer template, the resulting Al NTs exhibit slightly reduced spacing, as demonstrated in Fig. 3(c). To further investigate the influence of Al NT spacing on the device's avalanche effect, we employed simulation methods. The results revealed that spacing between Al NTs ranging from 10 to 80 nm can induce resonance effects. Furthermore, the lightning-rod effect, as discussed earlier, initiates localized avalanches, and the field enhancement caused by resonance between Al NTs further expands the localized avalanche region.

In the future, we plan to explore additional methods to prepare PS bead templates with smaller spacing. While this remains a challenging task, we remain optimistic that, through continuous efforts and accumulated experience, it will eventually be achievable.

3. I have a particular issue with Fig 2c and 2d. The specific detectivity was invented to express the quality of a specific bulk material platform for photodetection, and cannot be freely applied to dimensionally-structured materials without specific caveats [Wang, F., Zhang, T., Xie, R. et al. How to characterize figures of merit of two-dimensional photodetectors. *Nat Commun* 14, 2224 (2023)]. The authors fail to mention what exact numbers were used to calculate the detectivity, in terms of responsivity, dark current and area. In particular, I suspect that the authors assumed an extremely small photosensitive area in their calculations, which is a conceptual mistake, as this is equivalent to discarding most of the incident light to be detected. In my opinion, the area used here should be the pitch of one square pixel in Fig. 1a. This will give a real estimation of how good the array is at detecting incident light (from the reference above: “for plane array devices, the entire device tiling the plane needs to be regarded as the optical area”). Otherwise, the authors may use a smaller area if they demonstrate a measurement of the performance of the array with MH array, which is capable of focusing the incident light onto an area smaller than the pixel size. I suggest that the authors specify the numbers used in their calculation, and add other performance metrics, such as internal and external quantum efficiency (the latter based on the total pixel area), which give a better indication of the useful performance of the detector.

This is particularly important as plasmonic enhancement is known to introduce additional loss [Khurgin, J. *How to deal with the loss in plasmonics and metamaterials. Nature Nanotech* 10, 2–6 (2015)], and it's necessary to estimate whether this strategy is worthwhile and provides a net improvement despite the plasmonic loss.

Fig. 2 (c) The corresponding detectivity with the wavelength range from 200 nm to 400 nm for the APDs with and without Al NTs at 10 V reverse bias. The inset shows detectivity from 260 nm to 300 nm.

Answer: Thank you for your comment. First, when measuring relatively low dark currents, there are lots of the fluctuations. At the 10 V point, the value just fluctuated below 1×10^{-14} A. We should consider the overall stable trend of the dark current. Therefore, we recalculated the detectivity based on the stable dark-current value, as shown in the above figure:

The specific expression for normalized detectivity is provided in the supplement $D^* = R \times \sqrt{\frac{A}{2qI_d}}$, which represents the photo-detection capability of the optoelectronic device and is correlated with the device's responsivity, dark current, and active area. In the "Design and Experiment" section of the article, the single square pixel is $200 \mu\text{m} \times 200 \mu\text{m}$ as shown in Figure 1(a), and the dark current I-V curve of the device is shown in Figure S4 in the supplementary material. Moreover, the spectral response is presented below and has already been incorporated into the supplementary material, which includes the device's responsivity and quantum efficiency. In summary, with the photosensitive area, dark current, and responsivity of the device, the detectivity can be obtained in Figures 2(c) and (d).

For example, for the 4 μm device at 280 nm at 10V, its responsivity R is 0.129 A/W, the dark current I_d is 5.0×10^{-14} A, and the device area is $200 \mu\text{m} \times 200 \mu\text{m}$. Thus the normalized detectivity D^* is 2.0×10^{13} Jones. The above-mentioned content has been presented in the main text and highlighted.

Fig. S6. Spectral response and quantum efficiency graphs of APD devices with MHs without Al NTs, with MHs with Al NTs, and without MHs with Al NTs, under 10 V reverse bias.

According to your comment, we have revised the main text of this section on **page 6** of the manuscript. The revised part is as follow:

“From the spectral response data, the 4 μm MH device without Al NTs showed peak responsivity of 0.133 A/W at 280 nm at 10 V (58.9% External Quantum Efficiency (EQE)) as shown in Fig. S6, while the Al NT-equipped 4 μm MH device reached 0.129 A/W (57.2% EQE). 42.8% of the incident photons do not contribute to the generation of photogenerated carriers. This proportion accounts for light reflection, carrier recombination due to defects, and field enhancement. Reflection spectrum testing, as shown in Fig. S13, indicates that the reflectivity at 280 nm is approximately 16.0%. Therefore, the photons contributing to field enhancement account for less than 26.8%. Similarly, 8 μm and 10 μm devices with Al NTs exhibited lower EQE values due to partial light reflection by Al NTs. Compared to 4 μm devices, the reduced responsivity and EQE in larger MH devices stem from incomplete depletion of lateral electric fields, lowering carrier collection efficiency. Under Al NT and MH-induced local avalanche, the 4 μm MH device achieved 3.35×10^{-4} A photocurrent with EQE of 2,374,473% at 50 V.

Furthermore, device detectivity was evaluated across wavelengths (Fig.2(c)). The 4 μm MH devices with Al NTs achieved peak detectivity of 2.0×10^{13} Jones at 280 nm under a reverse bias of 10 V.”

4. The authors should provide a clearer explanation of why the presented architecture allows to avoid the need for quenching circuits: is this because the devices are operated at sub-breakdown voltages? If this is the case, what is the advantage compared to operating conventional detectors at sub-breakdown voltages (say devices without Al NT at 50 V)? The sentence in lines 113-115 “The reduction in avalanche voltage offers

significant advantages, particularly in preventing instantaneous breakdown and safeguarding the device from catastrophic damage” is unclear or misleading: if the avalanche voltage is lower, it is typically easier to trigger breakdown and not harder.

Answer: Thank you for your insightful comment. The detection of individual photons, which inherently produce weak signals, necessitates the use of APD devices operating in Geiger mode. This mode involves applying a high voltage to create an internal electric field near the bulk avalanche breakdown threshold, thereby achieving significant gain. However, the detectors in Geiger mode requires a quenching circuit to dissipate the high electric field to prevent breakdown.

In this study, high gain has been achieved in the detectors through stable and repeatable local avalanche at lower voltages avoiding devastating bulk avalanche breakdown. This approach not only significantly reduces the avalanche voltage but also protects the devices from breakdown, which eliminate eliminating the need for a quenching circuit. The long-term stability and repeatability of the devices are demonstrated in Fig. S5.

According to your comment, we have revised the main text of this section on **page 7** of the manuscript. The revised part is as follow:

“In this study, high gain has been achieved in the detectors through stable and repeatable local avalanche at lower voltages avoiding devastating bulk avalanche breakdown. This approach not only significantly reduces the avalanche voltage but also protects the devices from breakdown, which eliminate the need for a quenching circuit. The long-term stability and repeatability of the devices are demonstrated in Fig. S5.”

5. the schematic in Fig. 4d is a simple representation of any avalanche process, without and specific relation to the presented devices (e.g. local avalanche, field enhancement), and I believe fails to capture the complex physics in action in these devices. In fact, the incident photons that provide a local field enhancement as shown in Fig. 3c do not contribute to the photo carrier generation. To investigate this, one should put a monitor plane in the FDTD simulation, situated a few 10s nm inside the substrate underneath the Al NT, and compare the transmitted number of photons vs the incident ones: this portion is the one contributing to photo carrier generation. The authors should implement and discuss this model.

Fig.4 (d) band diagram of the APD with MHs and Al NTs under the condition of illumination.

Fig. S11. Simulation diagrams of the electric field inside the device under different inter spacing(10 nm、 20 nm、 30 nm and 80 nm) conditions of Al NTs by COMSOL. When the inter-spacing between Al NTs is 10 nm, 20 nm, 30 nm, 80 nm, and for the device without Al NTs, the maximum electric fields within the devices are 4.84 MV/cm, 3.85 MV/cm, 3.66 MV/cm, 1.64 MV/cm, and 1.06 MV/cm, respectively.

Answer: Thank you for your comment. Below Figure 4(d) illustrates a schematic of the localized avalanche phenomenon, where it can be observed that carrier collision ionization occurs on both conduction band and valence band. This indicates that avalanche takes place at the edges of the MHs, whereas the remaining regions remain quiet. Figure S11 depicts the effect of varying inter-spacing between Al NTs on the

internal electric field of the device simulated by COMSOL. On one hand, it is evident that devices with Al NTs exhibit a significantly stronger internal electric field compared to those without Al, with the maximum field strength reaching 4.84 MV/cm at a spacing of 10 nm. Additionally, it is observed that the maximum internal electric field within the device increase as the spacing between Al NTs decrease. This increase is primarily due to the enhanced coupling effect that occurs between closely spaced Al NTs, which results in a stronger electric field.

Fig. S12. (a) The distributions of hole and (b) electron concentration within the device at 10V reverse bias.

Fig. S13. The reflection spectrum of the device with MH and Al NTs.

To better comprehend the carrier concentration within the device, we simulated the spatial distributions of hole and electron concentrations, as illustrated in Fig. S12. The results reveal that in regions with high electric fields, the depletion of the electric field is more pronounced, leading to an increased electron concentration. Furthermore, the contribution of incident photons to carrier generation and field enhancement can be analyzed through photocurrent and quantum efficiency. To elaborate, the photocurrent I_{ph} is measured as $I_{ph}=7.6 \times 10^{-9}$ A (@10 V/280 nm/ 5.89×10^{-8} W), with

the elementary charge $q = 1.6 \times 10^{-19}$ C, and a quantum efficiency $\eta = 57.2\%$. Based on these parameters, we can perform the following calculations. First, we calculate the number of photogenerated carriers produced within 1 second: $N = I_{ph} * t / q = 4.75 \times 10^{10}$, which corresponds to 47.5 billion electron-hole pairs. Subsequently, we determine the number of incident photons M , where $M = N / \eta = 8.3 \times 10^{10}$. In summary, 42.8% of the incident photons do not contribute to the generation of photogenerated carriers. This proportion accounts for light reflection, carrier recombination due to defects, and field enhancement. Reflection spectrum testing, as shown in Fig. S13, indicates that the reflectivity at 280 nm is approximately 16.0%. Therefore, the photons contributing to field enhancement account for less than 26.8%, corresponding to fewer than 2.3×10^{10} photons.

According to your comment, we have revised the main text of this section on **page 11** of the manuscript. The revised part is as follow:

“To further investigate the internal electric field distribution, we conducted COMSOL simulations on devices with varying Al NTs spacings, as illustrated in Fig. S11. The simulation results demonstrate unequivocally that devices incorporating Al NTs produce significantly stronger internal electric fields compared to Al-free devices, achieving a maximum field strength of 4.84 MV/cm at a spacing of 10 nm. Furthermore, the simulations reveal an inverse correlation between Al NT spacing and the maximum internal field strength. As the spacing increases, the field intensity diminishes. This phenomenon can be attributed to the enhanced coupling effects observed in closely spaced Al NTs, which serve to amplify the local electric field. These findings conclusively demonstrate the profound impact of Al NT spacing on device performance.”

According to your comment, we have revised the main text of this section on **page 6** of the manuscript. The revised part is as follow:

“42.8% of the incident photons do not contribute to the generation of photogenerated carriers. This proportion accounts for light reflection, carrier recombination due to defects, and field enhancement. Reflection spectrum testing, as shown in Fig. S13, indicates that the reflectivity at 280 nm is approximately 16.0%. Therefore, the photons contributing to field enhancement account for less than 26.8%.”

6. The sentence at lines 217-220 “This increase is attributed to the acceleration of photogenerated carriers within the space-charge region at higher reverse bias, necessitating a longer response time. Moreover, both T_r and T_f in devices with Al NTs increased by approximately 10% compared to devices without Al NTs” sounds counter-intuitive, as the higher voltage will accelerate charges faster. In fact, I suspect this increase in response time is due to the higher current changing the dynamic resistance and hence the RC of the device, as evidenced by the fact that the same increase is also registered for devices with Al NT at the same voltages. If the authors disagree, they should provide an objective argument for their claims.

Answer: Thank you for your insightful feedback. We agree with your observations and would like to provide more detailed explanation.

From the spectral response diagram, it is evident that under a reverse bias of 10 V,

the device incorporating Al exhibits higher light reflection compared to the device without Al. This increased reflection leads to a reduction in photocurrent. Consequently, we can infer that the dynamic resistance of the Al-based device is higher. According to the frequency formula for an RC oscillation circuit, $f = 1/2\pi RC$, the frequency f of the device with Al is lower, which results in a longer response time $T = 1/f$.

Furthermore, the overall capacitance of the device plays a crucial role in determining its response speed. The device with Al introduces an additional parallel capacitance between the device and the Al nanoparticles, which increases the overall capacitance compared to the device without Al. This increase in capacitance further contributes to a longer response time.

Considering the both factors—higher dynamic resistance and increased capacitance—we can comprehensively explain why the device with Al exhibits longer response times.

According to your comment, we have revised the main text of this section on **page 12** of the manuscript. The revised part is as follow:

“First, from the spectral response data, we observe that under 10 V reverse bias, the Al-containing device exhibits greater light reflection, resulting in lower photocurrent compared to the Al-free device. This suggests higher dynamic resistance in the Al-containing device. Following the RC oscillation circuit frequency formula ($f = 1/2\pi RC$), the reduced frequency (f) in Al-containing devices consequently yields longer response times ($T = 1/f$). Additionally, the frequency calculation reveals that total device capacitance impacts response speed. The Al-containing device demonstrates additional parallel capacitance between the device and Al nanoparticles relative to Al-free devices, increasing overall capacitance and consequently response time. Together, these factors account for the prolonged response times observed in Al-containing devices. Additionally, response speed may be influenced by the transit time of carriers, the drift current and the diffusion current within the device.”

7. finally, I believe the quality and presentation of the figures graphics can be significantly improved: for starter, the largest fonts in several figures are around ten times larger than the smallest in the same figures. Fig. 1a and 1b lack scale bars, and some of the numbers in Fig. 3 are not explained in the caption. The caption of Fig 2b mentions “statistical comparison”, but no statistics is shown or mentioned in the figure itself: I guess the bar plots are the average of some number of devices, but can the authors specify how many devices, how is this average obtained and insert the corresponding error bars? These are the minimum necessary improvements, but there are several more that can be implemented.

Answer: Thank you for your comment. We have revised all the figure legends according to your request. To be honest, due to the inability to perfectly control the uniformity of the PS (polystyrene) sphere template arrangement, there may be some variability in the size and spacing of the Al NTs on each device's surface. Therefore, while each device experiences a local avalanche effect, not many avalanche devices can achieve a gain of 10^4 . For devices with different MH sizes, we tested the photocurrent

and dark current of over 20 individual devices, and an error analysis histogram is presented in Fig. 2(b). It can be observed that not all 4 μm devices exhibit an avalanche onset voltage as low as 14.5 V, but they are all below the avalanche voltage of traditional APDs. This is primarily due to the non-uniform distribution of Al NTs. Similarly, the gain of the devices varies from 10^3 to 10^4 . In the future, we will further optimize the technology to achieve better uniformity.

According to your comment, we have revised the main text of this section on **page 6** of the manuscript. The revised part is as follow:

“We also tested the dark currents and photocurrents of several devices with 4 μm , 8 μm and 10 μm MHs incorporating Al NTs and calculated their avalanche onset voltages and maximum avalanche gains, with error analysis histograms shown in Fig. 2(b). The 4 μm MH devices exhibited smaller onset voltage fluctuations overall, while 8 μm and 10 μm devices showed larger fluctuations, primarily related to Al NT distribution within the MHs and varying degrees of local avalanche enhancement. Additionally, despite gain fluctuations, 4 μm MH devices consistently achieved higher gains than 8 μm and 10 μm devices, reaching up to 10,556 times, mainly because their lateral region could be fully depleted, creating larger local avalanche regions.”

Fig. 2 (b) Histograms of error analysis for the avalanche breakdown voltage (V_a) and gain of APDs with Al NTs and MHs of different diameters (4 μm , 8 μm , and 10 μm).

Dear Reviewers:

Thank you for your letter and the reviewers' comments on our submitted manuscript entitled "**Local Avalanche Photodetectors Driven by Lightning-rod Effect and Surface Plasmon Excitations**". These comments are very constructive and helpful to our paper and future research. We have studied comments carefully and have made corrections which we hope meet with approval. The detail responses to the reviewers' comments are as follows:

Reviewer #1 (Remarks to the Author):

Local Avalanche Photodetectors Driven by the Lightning-Rod Effect and Surface Plasmon Excitations by Fu et al.

The authors have comprehensively addressed all the concerns raised by the reviewers. In light of this, the manuscript may be accepted for publication.

Answer: Thanks for your comments. We are delighted to receive this news and sincerely appreciate the time and effort spent by you and the reviewers in evaluating our work.

Reviewer #2 (Remarks to the Author):

I commend the efforts of the authors and recognize that the quality of the manuscript has substantially improved. However, there is still a key unsolved issue, which I believe to be fundamental to the validity of the manuscript:

1.the main issue is with the answer to point 3 of reviewer#2 which is incomplete at best: the calculations shown are still missing a key factor, which is the incident light power that is used in calculating the responsivity values. Just extracting from the answer to point 5, 58.9 nW is used: is this power entirely focused inside the 4- μm MH or over the entire 200x200 μm pixel? I suspect that this value is obtained from the light incident over a large area of a few mm beam size, normalized to the area of the 4- μm MH ($12.6 \cdot 10^{-12} \text{ m}^2$), which is consistent with this number being later used for the EQE calculation. This is a wrong procedure because this area is not the same utilized in the detectivity calculation (i.e. 200x200 μm^2 , as mentioned by the authors). Since the authors do not provide any details on the experimental setup, it is hard to infer what numbers were used in the calculations. The best practice would be to: 1. Describe the setup (e.g. "a source with ___ mW total power was focused on a ___ m^2 area, resulting in a power ___ W incident over the 200x200 μm^2 pixel area "); 2. Report the numbers used for both the responsivity and the detectivity calculations.

Answer: Thanks for your comments. In the experiment, a light source with 5.89×10^{-8} W total power(@280 nm) was focused on a circular area with a diameter of approximately 120 μm , resulting in a power 5.89×10^{-8} W incident over the 200 μm \times 200 μm pixel area.

Figure. S15. (a) shows a schematic diagram of the device performance measurement system described in the supplementary. The system primarily consists of

two integrated parts: the optical path and the electrical circuit. The optical path, which generates and conditions the test light, is composed of a xenon lamp, an optical fibre, and a monochromator. The electrical measurement section is contained within an electromagnetic shielding box and incorporates a probe station connected to a 4200A-SCS semiconductor parameter analyser. The current-voltage (I-V) characteristics of the semiconductor devices were measured using the source measure units (SMUs) of the analyser.

Fig. S15. (a) Schematic diagram of the device performance measurement system. (b) Focused light spot on the device photosensitive area (optical microscope image)

Prior to device testing, the intensity of the ultraviolet light emitted from the xenon lamp was calibrated. This calibration was performed by integrating a UV-enhanced Si-222 standard photodetector into the system. The UV light signal, after being conditioned by an adjustment objective lens, was focused onto the active surface of the standard photodetector to generate a photocurrent. The calibrated optical power values across the UV wavelength range were then calculated from the known spectral responsivity of the standard detector.

For instance, at a wavelength of 280 nm, the Si-222 standard photodetector exhibits a spectral responsivity of 0.0958 A/W. With an applied bias, a photocurrent of 5.64×10^{-9} A was measured, yielding a calibrated optical power of 5.89×10^{-8} W. Subsequently, the fabricated devices were tested. As shown in Figure. S15. (b), the light from the xenon lamp was coupled through the optical fibre and focused into a circular spot with a diameter of approximately 120 μm. This confirms that the total optical power of 58.9 nW was illuminated onto the photosensitive area of the device. After aligning the light spot, the monochromator was set to a wavelength of 280 nm for the measurement, from which the photocurrent-voltage (I-V) curve of the device was obtained. Thus, for the 4 μm device at 280 nm at 10 V, its responsivity R is 0.129 A/W, the dark current I_d is 5.0×10^{-14} A, and the device area is 200 μm × 200 μm. Thus the normalized detectivity D^* is 2.0×10^{13} Jones. The specific calculation procedure is demonstrated by the following formula:

$$R = \frac{I_{ph}}{P_{opt}} = \frac{7.6 \times 10^{-9} \text{ A}}{5.89 \times 10^{-8} \text{ W}} = 0.129 \text{ A/W}$$

$$D^* = R \times \sqrt{\frac{A}{2qI_d}} = 0.129 \text{ A/W} \cdot \sqrt{\frac{0.02 \text{ cm} \cdot 0.02 \text{ cm}}{2 \times 1.6 \times 10^{-19} \text{ C} \cdot 5.0 \times 10^{-14} \text{ A}}} = 2.0 \times 10^{13} \text{ Jones}$$

According to your comment, we have revised the content of this section on **page 11** and **page 12** of the supplementary files. The revised part is as follow:

“In the experiment, a light source with 5.89×10^{-8} W total power(@280 nm) was focused on a circular area with a diameter of approximately 120 μm , resulting in a power 5.89×10^{-8} W incident over the 200 $\mu\text{m} \times 200 \mu\text{m}$ pixel area.

Figure. S15. (a) shows a schematic diagram of the device performance measurement system described in the manuscript. The system primarily consists of two integrated parts: the optical path and the electrical circuit. The optical path, which generates and conditions the test light, is composed of a xenon lamp, an optical fibre, and a monochromator. The electrical measurement section is contained within an electromagnetic shielding box and incorporates a probe station connected to a 4200A-SCS semiconductor parameter analyser. The current-voltage (I-V) characteristics of the semiconductor devices were measured using the source measure units (SMUs) of the analyser.

Prior to device testing, the intensity of the ultraviolet light emitted from the xenon lamp was calibrated. This calibration was performed by integrating a UV-enhanced Si-222 standard photodetector into the system. The UV light signal, after being conditioned by an adjustment objective lens, was focused onto the active surface of the standard photodetector to generate a photocurrent. The calibrated optical power values across the UV wavelength range were then calculated from the known spectral responsivity of the standard detector.

For instance, at a wavelength of 280 nm, the Si-222 standard photodetector exhibits a spectral responsivity of 0.0958 A/W. With an applied bias, a photocurrent of 5.64×10^{-9} A was measured, yielding a calibrated optical power of 5.89×10^{-8} W. Subsequently, the fabricated devices were tested. As shown in Figure. S15. (b), the light from the xenon lamp was coupled through the optical fibre and focused into a circular spot with a diameter of approximately 120 μm . This confirms that the total optical power of 58.9 nW was illuminated onto the photosensitive area of the device. After aligning the light spot, the monochromator was set to a wavelength of 280 nm for the measurement, from which the photocurrent-voltage (I-V) curve of the device was obtained.”

Other comments:

2、 *the discussion and calculations with specific numbers used in the answers to point 2 of reviewer#1 and point 3 and 5 of reviewer#2 should be added to the supplementary*

Answer: Thanks for your comments. Based on your comments, we have added the corresponding content to the supplementary files on **page 4**, **page 5**, **page 6**, **page 9**, **page 10** respectively. The revised part is as follow:

point 2 of reviewer#1

“The avalanche noise level depends on the ionization rate ratio (α_N/α_P) and the

avalanche gain (M). For the case of hole injection, the noise level can be expressed as :

$$F=M[1-(1-k)\left(\frac{M-1}{M}\right)^2]$$

$$\approx kM+(2-\frac{1}{M})(1-k)$$

where $k=\alpha_N/\alpha_P$, which remains constant throughout the avalanche region. For SiC material, the impact ionization coefficients of electrons and holes can be expressed as:

$$\alpha_N(E)=1.69\times 10^6 \text{ cm}^{-1}\exp\left[-\left(\frac{9.96\times 10^6 \text{ V/cm}}{E}\right)^{1.6}\right] \quad (\text{electron})$$

$$\alpha_P(E)=3.32\times 10^6 \text{ cm}^{-1}\exp\left[-\left(\frac{1.07\times 10^7 \text{ V/cm}}{E}\right)^{1.1}\right] \quad (\text{hole})$$

where E represents the electric field intensity.

The avalanche electric field intensity in conventional SiC APDs is typically 3 MV/cm. By substituting $E=3 \text{ MV/cm}$ into the equations above, we calculate that $k \approx 0.032$. Consequently, the noise level in conventional devices can be expressed as:

$$F=0.032M-0.968/M+1.936$$

For comparison, the local avalanche electric field intensity in our fabricated device is 1.5 MV/cm under a reverse bias of 15 V as shown in Fig. S8. We therefore simulated the local avalanche electric field intensity under different reverse bias conditions. Since the gain is related to the reverse bias (as shown in Fig. 2a), Fig. S7 was obtained by curve fitting.

The results show that the noise level increases with avalanche gain. The fabricated device exhibits a lower noise level compared to the device without MH-Al NTs, owing to localized avalanche effects.” on **page 5** and **page 6** in the supplementary.

point 3 of reviewer#2

“The detectivity of the device shown in Figure 3(c) of the main text is calculated based on the responsivity from the Supplementary Material's Fig. S6 and the dark current from Fig. S4, with the specific calculation method and numerical discussion as follows:

The specific expression for normalized detectivity is provided in the supplement $D^*=R\times\sqrt{\frac{A}{2qI_d}}$, which represents the photo-detection capability of the optoelectronic device and is correlated with the device's responsivity, dark current, and active area. In the "Design and Experiment" section of the article, the single square pixel is $200 \mu\text{m} \times 200 \mu\text{m}$ as shown in Figure 1(a), and the dark current I-V curve of the device is shown in Figure S4 in the supplementary material. Moreover, the spectral response is presented below and has already been incorporated into the supplementary material, which includes the device's responsivity and quantum

efficiency. In summary, with the photosensitive area, dark current, and responsivity of the device, the detectivity can be obtained in Figures 2(c) and (d).

For example, for the 4 μm device at 280 nm at 10 V, its responsivity R is 0.129 A/W, the dark current I_d is 5.0×10^{-14} A, and the device area is $200 \mu\text{m} \times 200 \mu\text{m}$. Thus the normalized detectivity D^* is 2.0×10^{13} Jones. The specific calculation procedure is demonstrated by the following formula:

$$R = \frac{I_{\text{ph}}}{P_{\text{opt}}} = \frac{7.6 \times 10^{-9} \text{ A}}{5.89 \times 10^{-8} \text{ W}} = 0.129 \text{ A/W}$$

$$D^* = R \times \sqrt{\frac{A}{2qI_d}} = 0.129 \text{ A/W} \cdot \sqrt{\frac{0.02 \text{ cm} \cdot 0.02 \text{ cm}}{2 \times 1.6 \times 10^{-19} \text{ C} \cdot 5.0 \times 10^{-14} \text{ A}}} = 2.0 \times 10^{13} \text{ Jones}''$$

On **page 4** and **page 5** in the supplementary.

point 5 of reviewer#2

“Below Figure 4(d) illustrates a schematic of the localized avalanche phenomenon, where it can be observed that carrier collision ionization occurs on both conduction band and valence band. This indicates that avalanche takes place at the edges of the MHs, whereas the remaining regions remain quiet. Figure S11 depicts the effect of varying inter-spacing between Al NTs on the internal electric field of the device simulated by COMSOL. On one hand, it is evident that devices with Al NTs exhibit a significantly stronger internal electric field compared to those without Al, with the maximum field strength reaching 4.84 MV/cm at a spacing of 10 nm. Additionally, it is observed that the maximum internal electric field within the device increase as the spacing between Al NTs decrease. This increase is primarily due to the enhanced coupling effect that occurs between closely spaced Al NTs, which results in a stronger electric field.

To better comprehend the carrier concentration within the device, we simulated the spatial distributions of hole and electron concentrations, as illustrated in Fig. S12. The results reveal that in regions with high electric fields, the depletion of the electric field is more pronounced, leading to an increased electron concentration. Furthermore, the contribution of incident photons to carrier generation and field enhancement can be analyzed through photocurrent and quantum efficiency. To elaborate, the photocurrent I_{ph} is measured as $I_{\text{ph}} = 7.6 \times 10^{-9}$ A (@10 V/280 nm/ 5.89×10^{-8} W), with the elementary charge $q = 1.6 \times 10^{-19}$ C, and a quantum efficiency $\eta = 57.2\%$. Based on these parameters, we can perform the following calculations. First, we calculate the number of photogenerated carriers produced within 1 second: $N = I_{\text{ph}} \cdot t / q = 4.75 \times 10^{10}$, which corresponds to 47.5 billion electron-hole pairs. Subsequently, we determine the number of incident photons M, where $M = N / \eta = 8.3 \times 10^{10}$. In summary, 42.8% of the incident photons do not contribute to the generation of photogenerated carriers. This proportion accounts for light reflection, carrier recombination due to defects, and field enhancement. Reflection spectrum testing, as shown in Fig. S13, indicates that the reflectivity at 280 nm is approximately 16.0%. Therefore, the photons contributing to field enhancement account for less than 26.8%, corresponding to fewer than 2.3×10^{10} photons.” on **page 9** and **page 10** in the

supplementary.

3、 the following paragraph on the statistics in figure 2(b) should be added to the text or supplementary “To be honest, due to the inability to perfectly control the uniformity of the PS (polystyrene) sphere template arrangement, there may be some variability in the size and spacing of the Al NTs on each device's surface. Therefore, while each device experiences a local avalanche effect, not many avalanche devices can achieve a gain of 10^4 . For devices with different MH sizes, we tested the photocurrent and dark current of over 20 individual devices, and an error analysis histogram is presented in Fig. 2(b). It can be observed that not all 4 μm devices exhibit an avalanche onset voltage as low as 14.5 V, but they are all below the avalanche voltage of traditional APDs. This is primarily due to the non-uniform distribution of Al NTs. Similarly, the gain of the devices varies from 10^3 to 10^4 .”

Answer: Thanks for your comments. We have added the following text on **page 6** in the manuscript as suggested. The revised part is as follows:

“However, to be honest, due to the inability to perfectly control the uniformity of the PS (polystyrene) sphere template arrangement, there may be some variability in the size and spacing of the Al NTs on each device's surface. Therefore, while each device experiences a local avalanche effect, not many avalanche devices can achieve a gain of 10^4 as described in Fig. 2(b). It can also be observed that not all 4 μm devices exhibit an avalanche onset voltage as low as 14.5 V, but they are all below the avalanche voltage of traditional APDs. This is primarily due to the non-uniform distribution of Al NTs. Similarly, the gain of the devices varies from 10^3 to 10^4 .”